# Asymmetric hydrogenation of 1,1-diarylethylenes and benzophenones through a relay strategy

Ke Li[1,3], Wen-Qiang Wu[1,3], Yunzhi Lin[1] & Hang Shi [1,2] ✉

Homogenous transition-metal catalysts bearing a chiral ligand are widely used for asymmetric hydrogenation of unsaturated compounds such as olefins and ketones, providing efficient concise access to products with chiral carbon centers. However, distinguishing the *re* and *si* prochiral faces of a double bond bearing two substituents that are sterically and electronically similar is challenging for these catalysts. Herein, we report a relay strategy for constructing compounds with a chiral *gem*-diaryl carbon center by means of a combination of selective arene exchange between 1,1-diarylethylenes or benzophenones with (naphthalene)Cr(CO)$_3$ and subsequent asymmetric hydrogenation. During the hydrogenation, the Cr(CO)$_3$ unit facilitate differentiation of the two prochiral faces of the substrate double bond via formation of a three-dimensional complex with one of the aromatic rings by selective arene exchange. Density functional theory calculations reveal that during the hydrogenation, chromium coordination affected π–π stacking of the substrate and the catalyst ligand, leading to differentiation of the prochiral faces.

Chiral carbon centers are present in a wide variety of functional molecules, including natural products, pharmaceuticals, and agrochemicals[1]. Extensive study of asymmetric hydrogenation reactions over the last several decades has led to the establishment of a number of remarkable methods involving homogeneous transition-metal catalysis, providing reliable access to chiral compounds from abundant unsaturated substrates[2–15]. To achieve high enantioselectivity, researchers have designed chiral catalysts that can distinguish between the two prochiral faces of substrates on the basis of differences between the substituents on the double bonds[16–29]. However, general methods for discriminating between the *re* and *si* faces of double bonds bearing two different aryl groups, which would provide compounds with chiral *gem*-diaryl motifs[30–41], are still under development owing to the small difference in electronic properties between the two aromatic rings and to their similar planar structures.

Depending on the positions of the substituents on the aromatic rings of 1,1-diaryl unsaturated compounds such as 1,1-diarylethylenes and benzophenones, they can be divided into two categories. High

enantioselectivities have been achieved for category a, compounds in which one of the aromatic rings bears an *ortho* substituent (Fig. 1a)[42–52]. For example, Wang et al. reported rhodium/Duanphos-catalyzed asymmetric hydrogenation of 1,1-diarylethylenes bearing a *ortho*-hydroxy directing group[44]. In addition, Song et al. accomplished elegant syntheses of enantiomerically enriched 1,1-diarylethanes by means of iridium-catalyzed hydrogenation reactions assisted by an *ortho* carboxylic acid group[45]. Sterically bulky substituents without a chelating atom can also facilitate catalyst differentiation of the prochiral faces of double bonds. For example, Mazuela et al. accomplished asymmetric hydrogenation of 1,1-diarylethylenes by using an iridium phosphite–oxazoline catalyst[42], and Chen et al. used a chiral oxazoline iminopyridine–cobalt catalyst for enantioselective hydrogenation of 1,1-diarylethylenes[48]. Ohkuma et al.[49] and Touge et al.[52] found that ruthenium-catalyzed asymmetric hydrogenation of benzophenones also requires an *ortho* substituent for high enantioselectivity. In addition, owing to the less difference of steric hindrance, chiral induction in hydrogenation of unsaturated compounds bearing two *ortho*-

[1]Key Laboratory of Precise Synthesis of Functional Molecules of Zhejiang Province, Department of Chemistry, School of Science and Research Center for Industries of the Future, Westlake University, 600 Dunyu Road, Hangzhou 310030, P. R. China. [2]Institute of Natural Sciences, Westlake Institute for Advanced Study, 18 Shilongshan Road, Hangzhou 310024, P. R. China. [3]These authors contributed equally: Ke Li, Wen-Qiang Wu. ✉e-mail: shihang@westlake.edu.cn

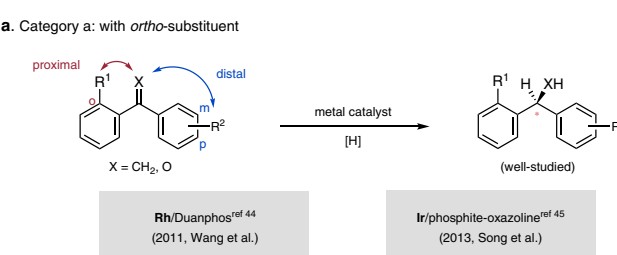

**a**. Category a: with *ortho*-substituent

**b**. Category b: without *ortho*-substituent (challenging)

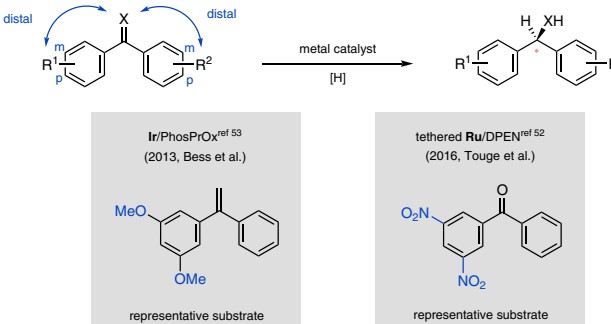

**Fig. 1 | Background of metal-catalyzed asymmetric hydrogenation of 1,1-diarylethylenes and benzophenones. a** Metal-catalyzed asymmetric hydrogenation of 1,1-diarylethylenes and benzophenones with *ortho*-substituent. **b** Metal-catalyzed asymmetric hydrogenation of 1,1-diarylethylenes and benzophenones without *ortho*-substituent.

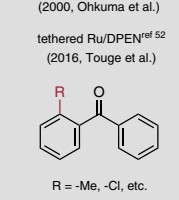

**Fig. 2 | Asymmetric reduction of chromium η⁶-arene complexes. a** Synthesis and Corey-Bakshi-Shibata (CBS) reduction of $Cr(CO)_3$ benzophenone complex. **b** Relative stability of $Cr(CO)_3$ arene complex. **c** This work: synthesis of compounds bearing a chiral *gem*-diaryl motif from either 1,1-diarylethylenes or benzophenones by a sequence involving selective arene exchange and stereoselective asymmetric hydrogenation.

substituted aromatic rings is challenging; a highly enantioselective catalytic method has yet to be reported.

In stark contrast, asymmetric hydrogenation of 1,1-diaryl-substituted double bonds without the assistance of a proximal substituent is fundamentally difficult, and enantioselectivities are generally low (Fig. 1b)[53–56]. Exceptionally, double bonds bearing highly electronically biased aromatic rings have been successfully hydrogenated with good enantioselectivities. For example, Bess et al. achieved asymmetric hydrogenation of 1,1-diarylalkenes bearing a 3,5-dimethoxylphenyl ring by using an iridium/phosphoramidite catalyst[53], and Touge et al. observed high enantioselectivities in ruthenium/1,2-diphenyl-1,2-ethylenediamine-catalyzed transfer hydrogenation of benzophenones with electron-poor aromatic rings, such as a 3,5-dinitrophenyl ring[52].

To provide a reliable method for asymmetric hydrogenation of 1,1-diaryl-substituted double bonds, we wondered a transition-metal unit would facilitate the differentiation of the two prochiral faces by ligating with one of the aromatic rings. Metal η⁶-benzene complexes, such as (benzene)$Cr(CO)_3$, have different electronic properties than free benzene and have three-dimensional structures[57–66]. In 1995, Corey and co-workers conducted an elegant study of the electronic effects of remote substituents on aromatic rings during Corey-Bakshi-Shibata (CBS) reduction of benzophenones and found that $Cr(CO)_3$-ligated *p*'-chlorobenzophenone synthesized from (benzene)$Cr(CO)_3$ can be reduced to the corresponding alcohol with 94% ee (Fig. 2a)[67–70]. Given that the stability of a metal η⁶-arene complex varies with the ligated aromatic ring (for an example involving chromium, see Fig. 2b)[71–73], we envisioned that combining selective arene coordination with a metal unit and transition-metal-catalyzed asymmetric hydrogenation would provide a reliable solution to the challenge posed by category b. Two questions needed to be answered: Could a metal unit distinguish and coordinate selectively to only one aromatic ring of a conjugated compound, and would the metal complexes be compatible with homogenous hydrogenation conditions[74,75]?

Herein, we report a relay strategy for synthesis of compounds bearing a chiral *gem*-diaryl motif from either 1,1-diarylethylenes or benzophenones by a sequence involving selective arene exchange with (naphthalene)$Cr(CO)_3$ (**2**) and subsequent ruthenium-catalyzed asymmetric hydrogenation (Fig. 2c).

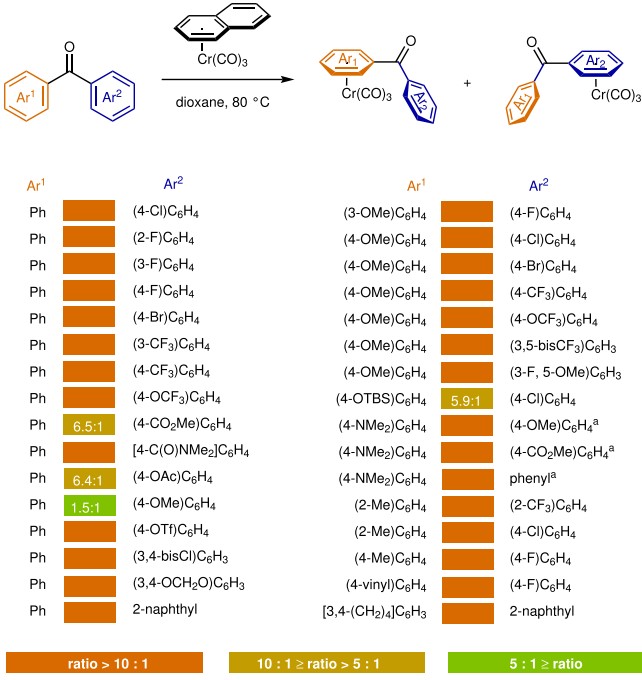

| Ar¹ | ratio | Ar² | Ar¹ | Ar² | ratio | Ar² |
|---|---|---|---|---|---|---|
| Ph | | (4-Cl)C₆H₄ | (3-OMe)C₆H₄ | (4-F)C₆H₄ | | |
| Ph | | (2-F)C₆H₄ | (4-OMe)C₆H₄ | (4-Cl)C₆H₄ | | |
| Ph | | (3-F)C₆H₄ | (4-OMe)C₆H₄ | (4-Br)C₆H₄ | | |
| Ph | | (4-F)C₆H₄ | (4-OMe)C₆H₄ | (4-CF₃)C₆H₄ | | |
| Ph | | (4-Br)C₆H₄ | (4-OMe)C₆H₄ | (4-OCF₃)C₆H₄ | | |
| Ph | | (3-CF₃)C₆H₄ | (4-OMe)C₆H₄ | (3,5-bisCF₃)C₆H₃ | | |
| Ph | | (4-CF₃)C₆H₄ | (4-OMe)C₆H₄ | (3-F, 5-OMe)C₆H₃ | | |
| Ph | | (4-OCF₃)C₆H₄ | (4-OTBS)C₆H₄ | | 5.9:1 | (4-Cl)C₆H₄ |
| Ph | 6.5:1 | (4-CO₂Me)C₆H₄ | (4-NMe₂)C₆H₄ | (4-OMe)C₆H₄ᵃ | | |
| Ph | | [4-C(O)NMe₂]C₆H₄ | (4-NMe₂)C₆H₄ | (4-CO₂Me)C₆H₄ᵃ | | |
| Ph | 6.4:1 | (4-OAc)C₆H₄ | (4-NMe₂)C₆H₄ | phenylᵃ | | |
| Ph | 1.5:1 | (4-OMe)C₆H₄ | (2-Me)C₆H₄ | (2-CF₃)C₆H₄ | | |
| Ph | | (4-OTf)C₆H₄ | (2-Me)C₆H₄ | (4-Cl)C₆H₄ | | |
| Ph | | (3,4-bisCl)C₆H₃ | (4-Me)C₆H₄ | (4-F)C₆H₄ | | |
| Ph | | (3,4-OCH₂O)C₆H₃ | (4-vinyl)C₆H₄ | (4-F)C₆H₄ | | |
| Ph | | 2-naphthyl | [3,4-(CH₂)₄]C₆H₃ | 2-naphthyl | | |

| ratio > 10 : 1 | 10 : 1 ≥ ratio > 5 : 1 | 5 : 1 ≥ ratio |
|---|---|---|

**Fig. 3 | Selective formation of Cr(CO)₃-benzophenone complex.** Arene exchange reactions between (naphthalene)Cr(CO)₃ (**2**) and an array of benzophenones. Reaction conditions, unless otherwise stated: benzophenones (0.10 mmol), **2** (0.15 mmol), dioxane (1.0 mL), N₂, 80 °C, 24 h. ᵃ**2** (0.11 mmol). Ratios were determined by ¹H NMR spectroscopy.

## Results

### Reaction optimization

To explore the feasibility of our strategy, we began by investigating arene exchange reactions between different metal species and 4-chlorobenzophenone (**1a**) as a model substrate (see Supplementary Fig. 1). To our delight, upon reaction of **1a** with (naphthalene)Cr(CO)₃ (**2**)[76–80] in dioxane at 80 °C, arene exchange took place selectively on the unsubstituted benzene ring (**1a-Cr: 1a-Cr'** > 10:1).

The arene exchange conditions showed great generality, allowing for ligation of an array of benzophenones with generally high regioselectivities (Fig. 3). Specifically, the selectivity for the two aromatic rings was in line with the trend in the relative stabilities of the Cr(CO)₃–arene complexes shown in Fig. 2b. In addition, we found that the selectivity varied somewhat with temperature. For example, after a mixture of Cr-ligated 4-methoxylbenzophenone (**1b**) was heated at 140 °C for 3 h, the **1b-Cr/1b-Cr'** ratio increased to 4.4:1 from the 1.5:1 ratio that was observed under our kinetically controlled conditions, and a complex bearing one Cr(CO)₃ units on each aromatic ring was detected (see Supplementary Fig. 3). We attributed this result to arene exchange between complexes under thermodynamic conditions, and metal migration along the conjugated system may also have taken place[81–83].

Next, we investigated the asymmetric hydrogenation of Cr(CO)₃-ligated **1a** as a model substrate by using various chiral Ru catalysts (see Supplementary Fig. 4a). To our delight, the Ru(II)-NHC-diamine catalyst developed by Glorius group[84] afforded the corresponding reduced complex with high enantioselectivity (93% ee) upon reaction with H₂ gas (5 atm) in the presence of NaOᵗBu. The free alcohol could be readily released from the chromium with no obvious loss of enantiomeric purity by irradiation of the crude reaction mixture at 440 nm under air[85]. In contrast, when **1a** was directly subjected to the same hydrogenation conditions, the enantioselectivity was much lower (43% ee) (see Supplementary Fig. 4b). Beside **Ru-1**, we also found that the spiro diphosphines supported ruthenium catalyst developed by Zhou

group[86] also provided alcohol **4a** with high enantioselectivity (91% ee) (see Supplementary Fig. 4a).

### Substrate scope

Having optimized the conditions for the selective arene exchange and the asymmetric hydrogenation, we evaluated the scope of the sequence by applying it to an array of benzophenones **1** (Fig. 4). Monosubstituted benzophenones were suitable substrates (**4a–4j**), providing the desired alcohols in moderate to good yields with high enantioselectivities, regardless of whether the substituent was in the *para* or *meta* position. In addition, substrates bearing bisfunctionalized aromatic rings (**4k–4u**) and polyfunctionalized aromatic rings (**4v, 4w**), were suitable and gave high enantioselectivities (up to 99% ee). Notably, an array of functional groups, such as carboxylate (**4h**), amide (**4i**), piperonyl (**4k**), and siloxy (**4q**), were well tolerated under the reaction conditions.

Next, we evaluated the utility of the sequence for asymmetric hydrogenation of several 1,1-diarylethylenes (Fig. 4). By using hexane as the solvent for hydrogenation instead of toluene, we could obtain 1,1-diarylethanes in good yields with moderate to high enantioselectivities (up to 99% ee); both mono- (**5a–5c**) and disubstituted (**5d**) 1,1-diarylethylenes were suitable substrates.

Complexation of an unsymmetrically 1,2- or 1,3-disubstituted arene ring to a metal via η⁶-coordination results in a molecule lacking symmetry elements[60]. To determine whether planar chirality affected the stereochemical control of the asymmetric hydrogenation, we subjected a 1:1 mixture of both enantiomers of Cr(CO)₃ complexes of 2-methyl-4′-chlorobenzophenone to the ruthenium-catalyzed hydrogenation conditions (Fig. 5a). This reaction yielded two pair of diastereomers with > 20:1 dr and 2.9:1 dr respectively. After removing the Cr(CO)₃ unit, we obtained the free alcohol (**4x**) in overall 98% yield and 74% ee, implying that chiral catalyst **Ru-1** dominated the differentiation of the two prochiral faces during the hydrogenation. Interestingly, shortening the reaction time to 15 minutes resulted in kinetic resolution of **1x-Cr**[87], providing alcohol **4x** in 92% ee.

Our strategy was also applicable to substrates that formed chromium complexes less selectively (Fig. 5b). For example, reaction of 4-methoxybenzophenone (**1b**) with (naphthalene)Cr(CO)₃ (**2**) gave a mixture of two isomers, **1b-Cr** and **1b-Cr'**, which could be separated. Each of the isomers was then converted to the same enantioenriched alcohol (**4b**) by reduction catalyzed by the catalyst **Ru-1** or its enantiomer **Ru-1'**, respectively.

In addition, we tested the versatility of our relay strategy by replacing H₂ gas with other easy-handle hydrogenation sources, and we found that asymmetric hydrogenation of Cr(CO)₃-ligated 4-chlorobenzophenone catalyzed by a 1,2-diphenyl-1,2-ethylenediamine-derived ruthenium complex **Ru-2**[88–90] with sodium formate provided alcohol **4a** with high enantioselectivity. Then we evaluated the substrate scope of the combination of arene exchange and asymmetric hydrogenation (Fig. 6). To our delight, an array of diaryl ketones were suitable for this alternative method, providing enantioenriched 1,1-diarylmethanols with HCO₂Na as the hydrogen source instead of H₂ gas. Notably, substrates with a vinyl group (**4ac**) were tolerated under the reaction conditions.

### Mechanistic studies

Next, we turned our attention to the origin of the high enantioselectivity achieved in the hydrogenation of Cr(CO)₃-ligated substrates. The steric bulk of the Cr(CO)₃ unit can be expected to have hindered rotation of the aromatic rings relative to the extent of rotation possible in the unbound precursor. Indeed, density functional theory calculations involving a relaxed scan of the dihedral angle formed by the carbonyl group and the aromatic ring connected to the carbonyl group were consistent with this expectation. The maximum barrier to rotation for each ring of 4-chlorobenzophenone (**1a**) was around

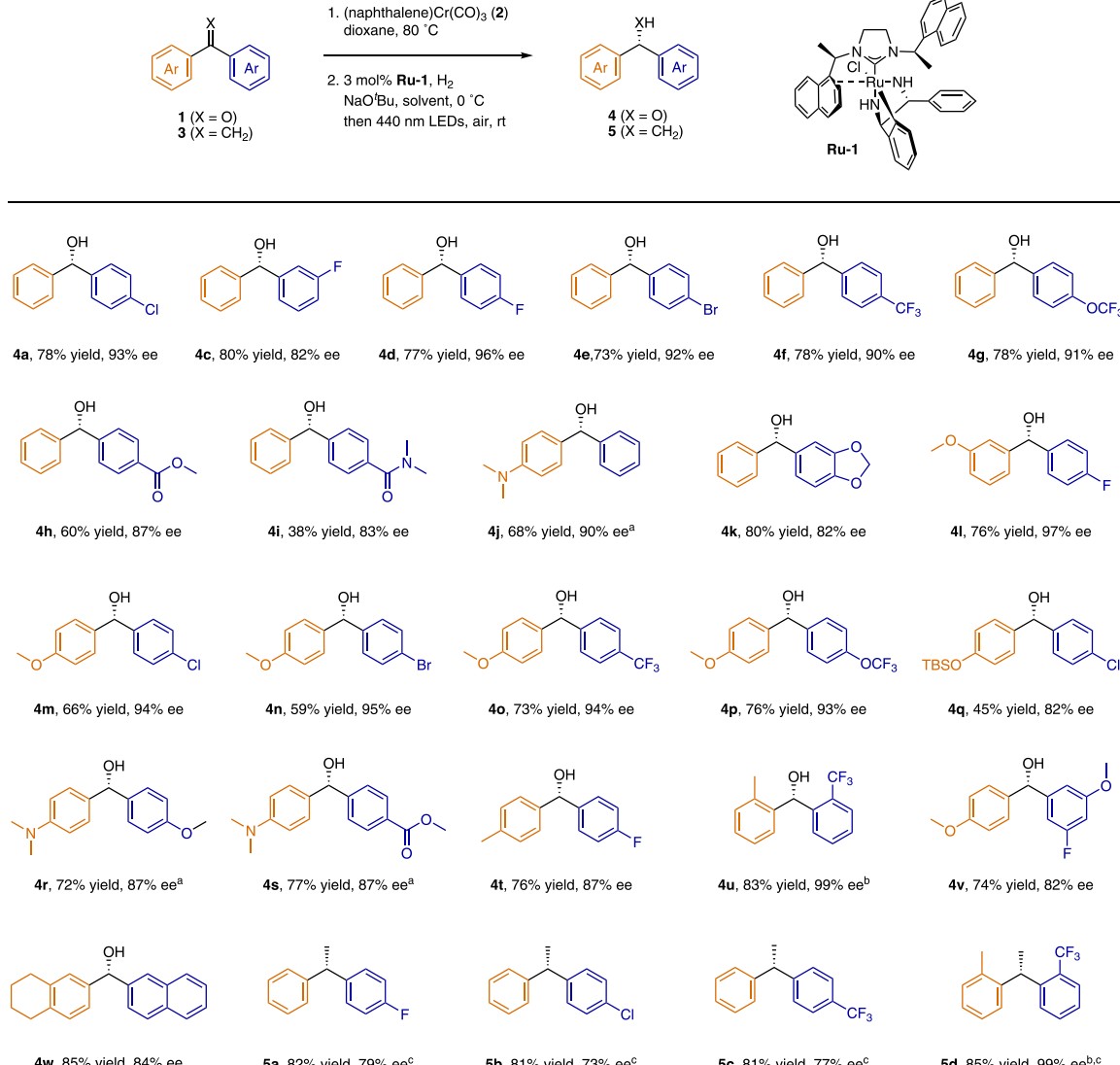

**Fig. 4 | Substrate scope of Ru-catalyzed hydrogenation reactions.** Synthesis of compounds bearing a chiral *gem*-diaryl motif from either 1,1-diarylethylenes or benzophenones by a sequence involving selective arene exchange and stereoselective asymmetric hydrogenation. Reaction conditions, unless otherwise stated: (1) **1** (0.2 mmol), **2** (0.3 mmol), dioxane (2 mL), 80 °C, 24 h. (2) **Ru-1** (3 mol%),

NaO*t*Bu (15 mol%), toluene (1.5 mL), H₂ (5 atm), 0 °C, 36 h; then irradiation with 440 nm LEDs for 2 h in air at room temperature. [a](1) **2** (0.22 mmol). [b](2) Room temperature. [c](2) Hexane (1.5 mL), H₂ (50 atm). Isolated yields were reported. The absolute configurations of **4a**, **4e**, **4 h**, and **4 m** were *S*, as determined by comparison of their optical rotations with reported data.

3 kcal/mol, and there was no obvious energetical difference between the *para*-Cl-substituted and unsubstituted rings (Fig. 7a). In contrast, the barrier to rotation of Cr(CO)₃-ligated phenyl group of **1a·Cr** was remarkably higher than that of the unbound phenyl ring, which may require a higher energy cost to adapt the benzophenone to the catalytic pocket (Fig. 7b).

To gain more insight into the mechanism of the stereoselectivity of the ruthenium-catalyzed hydrogenation, we performed additional density functional theory calculations. On the basis of previous studies[52,84], we began by investigating the reaction of noncoordinated 4-chlorobenzophenone (Fig. 8, left part), and we found that the transition states for the different stereoselectivities were quite similar in structure; there was only a tiny energy difference (0.3 kcal/mol) between them, which is consistent with the observed absolute stereochemical outcome and the low enantioselectivity (43% ee). Moreover, similar weak interactions (the green area, see IGMH analysis) between the aromatic substituents of the catalyst and the 4-chlorobenzophenone ring in transition states **TS-1** and **TS-2** lead to quite close in energy.

In contrast, calculations on the asymmetric hydrogenation of Cr(CO)₃-ligated 4-chlorobenzophenone revealed relatively lower energy barriers to the formation of both enantiomeric products, which can probably be attributed to the electron-withdrawing effect of the Cr(CO)₃ unit (for atomic charge population and conceptual density functional analysis see Supplementary Tables 2 and 3). Furthermore, transition state **TS-4**, formed by reduction from the *re* face, was favored by 2.1 kcal/mol over **TS-3**, formed by reduction from the *si* face (Fig. 8, right part). Specifically, in **TS-3**, the chromium-ligated benzene ring was located in a rigid pocket formed by a phenyl group of the diamine ligand and a naphthyl group of the carbene ligand, leading not only to greater steric repulsion (for activation strain model, see Supplementary Table 2), but also to a weaker π−π stacking interaction between the phenyl ring of the diamine ligand and the chromium-ligated benzene ring (the corresponding green area in **TS-3** was smaller than that in **TS-4**, see IGMH analysis) because rotation of the latter was hindered, as indicated by the clearly smaller |∠O1-C1C2-C3| angle (27.7°) relative to other transition states **TS-1** (39.2°), **TS-2** (39.1°), and **TS-4** (41.9°).

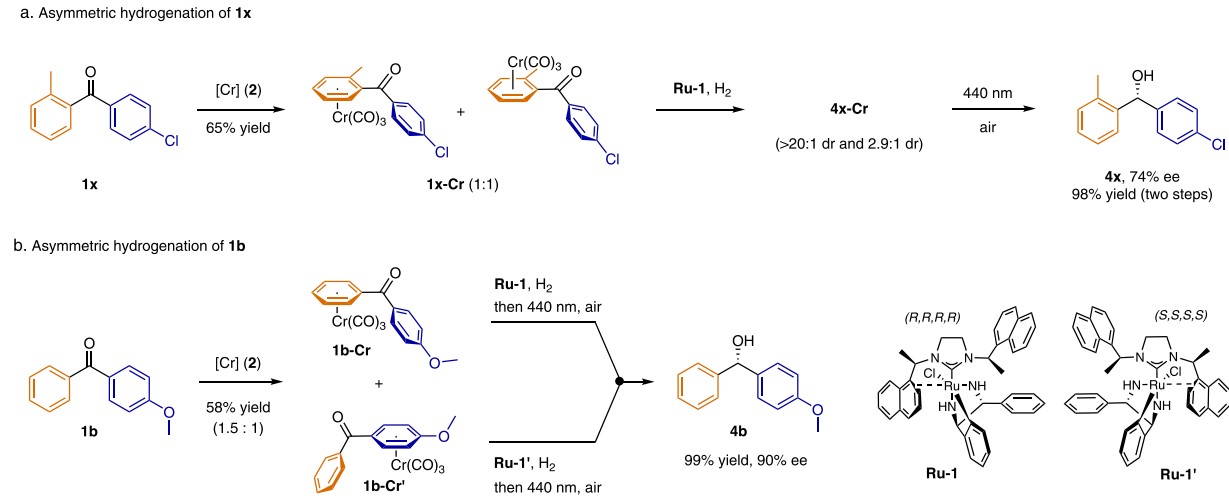

**Fig. 5 | Asymmetric hydrogenation of benzophenones 1x and 1b. a** Synthesis of **4x** by a sequence involving selective arene exchange and stereoselective asymmetric transfer hydrogenation. **b** Synthesis of **4b** by a sequence involving selective arene exchange and stereoselective asymmetric transfer hydrogenation.

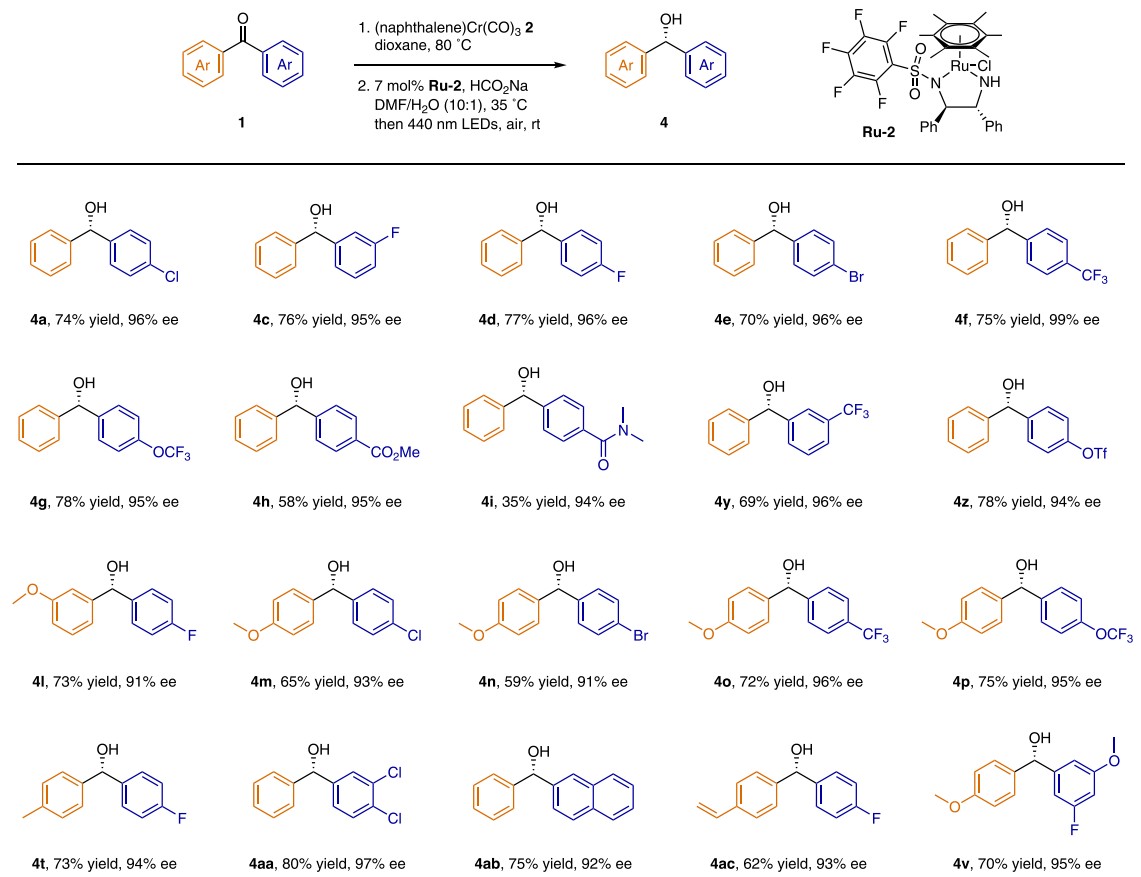

**Fig. 6 | Substrate scope of Ru-catalyzed transfer hydrogenation reactions.** Synthesis of compounds bearing a chiral *gem*-diaryl motif from benzophenones by a sequence involving selective arene exchange and stereoselective asymmetric transfer hydrogenation. Reaction conditions, unless otherwise noted: (1) **1** (0.2 mmol), **2** (0.3 mmol), dioxane (2.0 mL), 80 °C, 24 h. (2) **Ru-2** (7 mol%), NaCO₂H (2 mmol), DMF (1.0 mL), H₂O (0.1 mL), 35 °C, 36 h; then irradiation for 2 h with 440 nm LEDs. Isolated yields were reported. The absolute configurations of **4a**, **4e**, **4 h** and **4 m** were *S*, as determined by comparing their optical rotations with reported data.

In conclusion, we have developed a relay strategy for accessing compounds with a chiral carbon center bearing two aromatic substituents by means of a combination of selective arene exchange with (naphthalene)Cr(CO)₃ and subsequent enantioselective hydrogenation mediated by a chiral ruthenium catalyst. Density functional theory calculations revealed that the chromium markedly affected the energy of the transition state for addition of the ruthenium hydride species to the carbonyl group. Work aimed at extending this strategy to related transformations is currently underway in our laboratory.

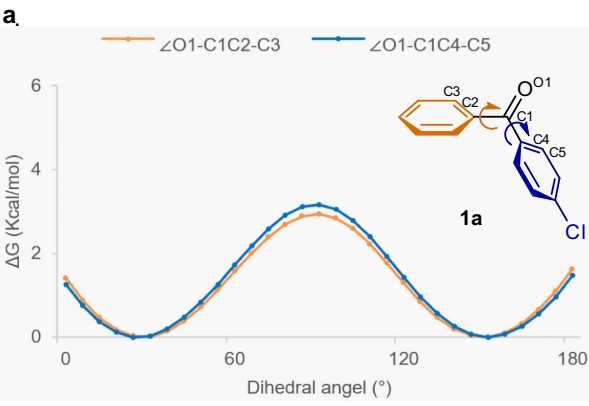

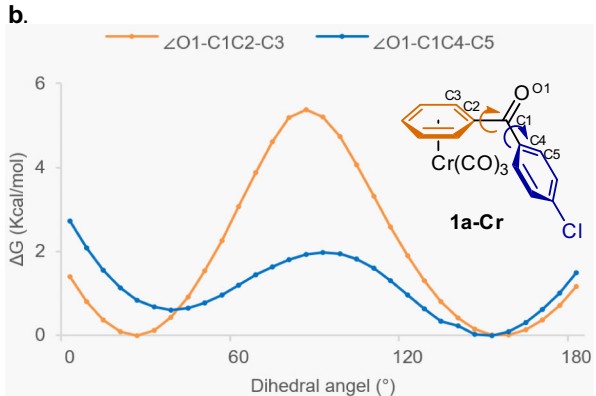

**Fig. 7 | Energy diagram of a relaxed scan of the dihedral angle between the carbonyl group and the aromatic ring in 1a and 1a-Cr.** The relative Gibbs free energies are given in kcal/mol. **a** Energy diagram of a relaxed scan of the dihedral angle between the carbonyl group and the aromatic ring in **1a**. **b** Energy diagram of a relaxed scan of the dihedral angle between the carbonyl group and the aromatic ring in **1a-Cr**.

**Fig. 8 | Computational studies of transition states for asymmetric hydrogenation of 1a and 1a-Cr.** The relative Gibbs free energies are given in kcal/mol. Left part, computational studies of transition states for asymmetric hydrogenation of **1a**. Right part, computational studies of transition states for asymmetric hydrogenation of **1a-Cr**.

## Methods

### General procedure for synthesis of Cr complexes

In a glovebox, a solution of Cr(CO)$_3$(naphthalene) **2** (0.3 mmol, 1.5 equiv.), **1** or **3** (0.2 mmol, 1.0 equiv.) in 1,4-dioxane (2 mL) in a 4 ml glass vial was stirred at room temperature for 2 min, then the sealed reaction vial was taken out of the glovebox and the reaction mixture was stirred at 80 °C in the dark for 24 h. After cooling to room temperature, the solution was evaporated under reduced pressure. The residue was purified by column chromatography eluting with Ethyl acetate/Petroleum ether, affording complex **1-Cr** or **3-Cr**.

### General procedure for asymmetric hydrogenation

(Figure 4) To a 5 ml glass tube, **Ru-1** (3 mol%), NaO$^t$Bu (15 mol%), **1-Cr** or **3-Cr** (1.0 equiv.), and toluene or hexane (0.10 M) were added in a glovebox. The glass tube was taken out of the glovebox and placed in a pre-cooled (0 °C) stainless steel autoclave, then the autoclave and pressurized and depressurized with hydrogen gas five times before the indicated pressure (5 atm or 50 atm) was set. The reaction mixture was stirred at 0 °C for 36 h. After the autoclave was carefully depressurized, the tube was irradiated under 440 nm LEDs (20 W) for 2 hours at room temperature in air atmosphere. Then the product was purified by flash column chromatography on silica gel. The enantiomeric excess was determined by SFC or HPLC analysis using chiral column. (Fig. 6) To a 4 ml glass vial, **Ru-2** (7 mol%), HCO$_2$Na (10 equiv.), **1-Cr** (1.0 equiv.), and DMF/H$_2$O (10:1, 0.18 M) were added. The reaction mixture was stirred at 35 °C for 36 h. Then the vial was irradiated under 440 nm LEDs (20 W) for 2 hours at room temperature in air atmosphere. Then the product was purified by flash column chromatography on silica gel. The enantiomeric excess was determined by SFC or HPLC analysis using chiral column.

## Data availability

Experimental procedures, characterization data, copies of NMR spectra and computational details are available in the Supplementary Information. Supplementary Data 1 contains the cartesian coordinates of the optimized structures. Supplementary Data 2 contains the detailed data of electronic energy of relaxed scan of dihedral angles. Data is available from the corresponding authors upon request.

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

## Acknowledgements
We thank the "Pioneer" and "Leading Goose" R&D Program of Zhejiang (2022SDXHDX0006), the National Natural Science Foundation of China (22071198, 22271235), and the Leading Innovative and Entrepreneur Team Introduction Program of Zhejiang (2020R01004) for research support. We thank Westlake university instrumentation and service center for molecular sciences and supercomputer center for the facility support and technical assistance.

## Author contributions
K.L. and W.-Q.W. contributed equally to this work. H.S. directed the project and wrote the manuscript. K.L. and W.-Q.W. performed the experiments and analysed the data. Y.L. performed density functional theory calculations. All authors discussed the results and commented on the manuscript.

## Competing interests
The authors declare no competing interests.
