## [Peer Review File · Nature Communications]

REVIEWER COMMENTS

Reviewer #1 (Remarks to the Author):

This manuscript describes a new approach toward the asymmetric reduction of unsaturated 1,1-diaryl compounds assisted by complexation with $\text{Cr}(\text{CO})_3$. It starts with a suitable introduction describing that enantioselective catalytic hydrogenation requiring precise discrimination of nonsymmetrical benzene rings of 1,1-diarylethylenes and benzophenones has been limited despite the fact that these are viable transformations for the construction of chiral geminal diaryl-substituted carbon centers. The authors commence with executing exchange reaction of a naphthalene ligand on Ru, Mn, Cr complexes with 4-chlorobenzophenone, followed by further investigation of the regioselectivity of the most promising $\text{Cr}(\text{CO})_3$. The selective arene exchange are then applied on a series of asymmetric hydrogenation and transfer hydrogenation of substituted benzophenones and 1,1-diarylethylenes, revealing that $\text{Cr}(\text{CO})_3$ -ligated substrates can facilitate chiral induction versatily for well-established chiral catalyst systems. Eventually, mechanistic aspects are discussed, based on DFT calculations for the combination of $\text{Cr}(\text{CO})_3$ ligation and the chiral Ru(II)-NHC-diamine catalyst. The experimental procedures are adequately described and the compounds are properly characterized. This paper is valuable to the synthetic community, and merits publication in Nature Communications after addressing the following issues adequately.

(1) Regarding the array of arene-Cr complexes in Fig. 2b, it would be better to compare the relative stability quantitatively. Put a validated parameter on the horizontal axis.

(2) Considering the fact that planar chirality of the $\text{Cr}(\text{CO})_3$ -ligated substrate affected the stereochemical results (as described in L. 126), an additional enhancement of optical purity of the product is to be achieved by performing kinetic resolution experiment. The authors are recommended to make further efforts to provide insightful data for the mechanism.

Reviewer #2 (Remarks to the Author):

In this manuscript, professor Shi and co-authors described an impressive relay method to achieve highly efficient asymmetric reduction of 1,1-diaryl ketones/olefins. Due to the similar steric sizes and electronic properties between aromatic substituent groups, reduction of Ar-Ar' ketones or alkenes remains a significant challenge in catalysis. Forming a η^6 -arene complexes with $\text{Cr}(\text{CO})_3$ enables the chiral catalysts to distinguish two aromatic rings in substrates, and thus realizing high

enantioselective reductions. Although this strategy requires stoichiometric amount of auxiliary $\text{Cr}(\text{CO})_3$, it does provide a useful way to obtain enantioenriched Ar-Ar' secondary alcohols. This is a useful contribution and could be published in Nat. Commun., but after addressing following issues.

- 1) How about asymmetric reduction of 2'-CF₃-benzophenone with this strategy?
- 2) Please supplement the HRMS-ESI data of $\text{Cr}(\text{CO})_3$ -benzophenone complex if these are unknown compounds. (S4-S10. i.e., S4, s1-Cr; S5, 1c-1e-Cr, 1y-Cr, 1f-1g-Cr; S6, 1h-1i-Cr, s2-Cr, 1b-Cr, 1b-Cr'...).
- 3) When flash chromatography is used for product purification, the solvents and their ratio should be identified for each compound.
- 4) Please re-purify the products (S125, 4d; S130, 4m; S131, 4n, 4o, 4p; S133, 4r; S138, 4x; S142, 5a; S143, 5b; S145, 5d.) to obtain clean ¹H NMR spectrum.

Reviewer #3 (Remarks to the Author):

In order to substantiate the obtained experimental results, the authors perform some DFT calculations to investigate the role of $\text{Cr}(\text{CO})_3$ coordination to benzophenone on the hydrogenation step. In particular, two facts are investigated: i) the hindered rotation of the phenyl group and, ii) the transition state for the ketone hydrogenation.

The computational level is adequate and the outcome of both calculations is meaningful. In Figure 4, it is shown that coordination of $\text{Cr}(\text{CO})_3$ to benzophenone increases the energetic barrier for the rotation of the Ph group. This explains its rigid character. In Figure 5, the presence of $\text{Cr}(\text{CO})_3$ leads to a stabilization of the TS-4 (5.7 kcal/mol) with respect to TS-3 (7.8 kcal/mol). An energetic difference of 2.1 kcal/mol explains the observed selectivity towards hydrogenation on re face. The energetic difference between TS-3 and TS-4 is discussed on the basis of steric repulsion and pi-pi stacking interactions using geometrical parameters. However, the arguments are rather qualitative. An analysis using the activation strain model could provide a quantitative insight.

In Figure 5, representations of "Independent gradient model based on Hirshfield partitioning" for TS(1-4) are shown, these figures are not discussed in the text. Such analysis could provide more detailed information between non-covalent interactions between the catalyst and the substrate and provide a sound explanation for the lower energetic barrier of TS-4. However, the obtained results are merely qualitative and do not show any relevant information.

Before acceptance of the manuscript, the authors should justify the claim that the effect of $\text{Cr}(\text{CO})_3$ on the benzophenone is electron-withdrawing. To do this, atomic charges for C and O atoms of benzophenone with and without $\text{Cr}(\text{CO})_3$ can be analyzed. The positive charge on the carbon atom should be increased upon coordination of $\text{Cr}(\text{CO})_3$.

In summary, the DFT study is very concise but the energetic data of the transition states provides very valuable information that explains, qualitatively, the experimental results.

Referee 1:

1. Comments: Regarding the array of arene-Cr complexes in Fig. 2b, it would be better to compare the relative stability quantitatively. Put a validated parameter on the horizontal axis.

Response: We thank the reviewer for this concern. The initial study of relative stability of (arene)Cr(CO)₃ complexes was demonstrated in the publication *J. Chem. Res., Synop.* **1979**, 126–127; however, we can not access the electronic version of the above paper. Therefore, we removed the horizontal axis and represented the relative stability of complexes qualitatively according to the previous paper on arene exchange (*J. Am. Chem. Soc.* **2005**, 127, 7759–7773). The following Fig. 2b was replenished in the revised manuscript.

b. Relative stability of Cr(CO)₃ arene complex^{ref 59-61}

2. Comments: Considering the fact that planar chirality of the Cr(CO)₃-ligated substrate affected the stereochemical results (as described in L. 126), an additional enhancement of optical purity of the product is to be achieved by performing kinetic resolution experiment. The authors are recommended to make further efforts to provide insightful data for the mechanism.

Response: We thank the reviewer for this concern. The kinetic resolution of **1x-Cr** can be achieved by shortening the reaction time. The enantioselectivity of alcohol **4x** was enhanced to 92% ee. The experiment has been replenished in the revised supplementary information as follows:

Supplementary Fig. 4 Kinetic resolution of **1x-Cr**. Conversion (C). S factor (S) = $\ln[(1 - C)(1 - ee_s)] / \ln[(1 - C)(1 + ee_s)]$.

The enantiomeric excess of recovered **1x-Cr** was determined by SFC: OD-3 column, MeOH/CO₂ = 5:95, 1.0 mL/min, 210 nm.

The enantiomeric excess of **4x** was determined by SFC: AD-3 column, MeOH/CO₂ = 10:90, 1.0 mL/min, 210 nm.

In addition, we have added a comment in the revised manuscript as follows:

“This reaction yielded two pair of diastereomers with > 20:1 dr and 2.9:1 dr respectively. After removing the Cr(CO)₃ unit, we obtained the free alcohol (**4x**) in overall 98% yield and 74% ee, implying that chiral catalyst **Ru-1** dominated the differentiation of the two prochiral faces during the hydrogenation. Interestingly, shortening the reaction time to 15 minutes resulted in kinetic resolution of **1x-Cr**, providing alcohol **4x** in 92% ee.”

Referee 2:

1. Comments: How about asymmetric reduction of 2'-CF₃-benzophenone with this strategy?

Response: We thank the reviewer for this concern. By using catalyst **Ru-1**, we evaluated our relay strategy in the hydrogenation of 2'-CF₃-benzophenone, and obtained the alcohol product in good yield with 96% ee.

The enantiomeric excess was determined by SFC: IG-3 column, MeOH/CO₂ = 10:90, 1.0 mL/min, 210 nm.

2. Comments: Please supplement the HRMS-ESI data of Cr(CO)₃-benzophenone complex if these are unknown compounds. (S4-S10. i.e., S4, s1-Cr; S5, 1c-1e-Cr, 1y-Cr, 1f-1g-Cr; S6, 1h-1i-Cr, s2-Cr, 1b-Cr, 1b-Cr' ...).

Response: We thank the reviewer for the suggestions. HRMS data of the Cr(CO)₃-benzophenone complexes have been replenished in the revised supplementary information.

3. Comments: When flash chromatography is used for product purification, the solvents and their ratio should be identified for each compound.

Response: We thank the reviewer for the suggestions. R^f and eluent information have been replenished in the revised supplementary information.

4. Comments: Please re-purify the products (S125, **4d**; S130, **4m**; S131, **4n**, **4o**, **4p**; S133, **4r**; S138, **4x**; S142, **5a**; S143, **5b**; S145, **5d**.) to obtain clean ^1H NMR spectrum.

Response: We thank the reviewer for the suggestions. The ^1H NMR spectrum of **4d** (Page S105), **4m** (Page S110), **4n** (Page S111), **4o** (Page S111), **4p** (Page S112), **4r** (Page S114), **4x** (Page S119), **5a** (Page S123), **5b** (Page S123) and **5d** (Page S125) have been updated in the revised supplementary information.

Referee 3:

1. Comments: The computational level is adequate and the outcome of both calculations is meaningful. In Figure 4, it is shown that coordination of $\text{Cr}(\text{CO})_3$ to benzophenone increases the energetic barrier for the rotation of the Ph group. This explains its rigid character. In Figure 5, the presence of $\text{Cr}(\text{CO})_3$ leads to a stabilization of the TS-4 (5.7 kcal/mol) with respect to TS-3 (7.8 kcal/mol). An energetic difference of 2.1 kcal/mol explains the observed selectivity towards hydrogenation on re face. The energetic difference between TS-3 and TS-4 is discussed on the basis of steric repulsion and pi-pi staking interactions using geometrical parameters. However, the arguments are rather qualitative. An analysis using the activation strain model could provide a quantitative insight.

Response: We thank the reviewer for the suggestions. The activation strain model is a tool to analyze activation barriers that determine reaction rates. For **TS-3** and **TS-4**, ΔE_{int} are approximately the same, and the distortion energies become the determining factor. It is easy for us to understanding this result, as the conformational changing of the rigid Ru catalyst was small in **TS-3** and **TS-4**, and the majority of activation strain could be attributed to the configurational diverse of benzophenones. As a result, the distortion energy of **TS-3** was larger than that of **TS-4**. However, the π - π stacking interactions are included in the ΔE_{int} , which is inseparable from the other two possible interaction (valance interaction and antibonding repulsion) and inhibit us from providing a quantitative insight. We have added a comment in the revised manuscript as follows:

“Specifically, in **TS-3**, the chromium-ligated benzene ring was located in a rigid pocket formed by a phenyl group of the diamine ligand and a naphthyl group of the carbene ligand, leading not only to greater steric repulsion (for activation strain model, see Supplementary Table 2),”

Supplementary Table 2 Distortion and interaction energies (kcal/mol) at PBE0-D3(BJ)/def2SVP level.

Entry	TS	$\Delta E_{\text{dist-sub}}$	$\Delta E_{\text{dist-cat}}$	ΔE_{dist}	ΔE_{int}
1	TS-3	9.0	3.8	12.8	-13.4
2	TS-4	6.9	2.5	9.4	-12.2

2. Comments: In Figure 5, representations of “Independent gradient model based on Hirshfeld partitioning” for TS(1-4) are shown, these figures are not discussed in the text. Such analysis could provide more detailed information between non-covalent interactions between the catalyst and the substrate and provide a sound explanation for the lower energetic barrier of TS-4. However, the obtained results are merely qualitative and do not shown any relevant information.

Response: We thank the reviewer for this concern. IGMH method was an update version of IGM method, which using the electron density gradient distribution in Hirshfeld partitioning in the real space to define interaction type and area. In this work we use IGMH method to define the weak interaction (π - π stacking) area and strength qualitatively. As shown in fig.5, we use δg isosurface colored by $\text{sign}(\lambda^2)\rho$ to give information of interaction between benzophenone(or corresponding $\text{Cr}(\text{CO})_3$ complex) and Ru catalysts in the relative hydrogenation transition states. Larger van der Waals interaction area of **TS-4** can be observed (the green area, from the π - π stacking between the phenyl ring of the diamine ligand and the 4-chlorophenyl ring of the **1a-Cr**, and other area is similar), which might lead to a lower TS free energy. However, just as the reviewer claimed, this method only provides a qualitatively result. To obtain the quantitative interaction information, SPAT energy separation calculation or EDA analysis could be conducted; but our computational resource is limited, and we haven't carried out such analysis. We have added comments in the revised manuscript as follows:

“Moreover, similar weak interactions (see IGMH analysis, the green area) between the aromatic substituents of the catalyst and the 4-chlorobenzophenone ring in transition states **TS-1** and **TS-2**

lead to quite close, which was support by the similar δg isosurface shape by density gradient distribution in Hirshfeld partitioning analysis.”

“but also to a weaker π - π stacking interaction between the phenyl ring of the **diamine** ligand and the chromium-ligated benzene ring (the corresponding green area in **TS-3** was smaller than that in **TS-4**, see IGMH analysis) because rotation of the latter was hindered, as indicated by the clearly smaller $|\angle O1-C1C2-C3|$ angle (27.7°) relative to other transition states **TS-1** (39.2°), **TS-2** (39.1°), and **TS-4** (41.9°).”

3. Comments: Before acceptance of the manuscript, the authors should justify the claim that the effect of $\text{Cr}(\text{CO})_3$ on the benzophenone is electron-withdrawing. To do this, atomic charges for C and O atoms of benzophenone with and without $\text{Cr}(\text{CO})_3$ can be analyzed. The positive charge on the carbon atom should be increased upon coordination of $\text{Cr}(\text{CO})_3$.

Response: We thank the reviewer for the suggestion. Atomic charge population was a powerful tool for providing quantities information for increased electron-withdrawing effect by $\text{Cr}(\text{CO})_3$ η^6 -coordination, we calculated various atomic charge population including Hirshfeld atomic charge, CM5 atomic charge and ADCH atomic charge (see Supplementary Table 3). Compared to the unbound arene, positive atomic charge of the complex was relatively higher. In addition, we further performed the conceptual density functional analysis (see Supplementary Table 4). The enhancements of both vertical electron affinity of **1a-Cr** and localized Fukui index f_+ of C1 also supported that $\text{Cr}(\text{CO})_3$ unit provided an electron withdrawing effect through η^6 -coordination. We have added a comment in the revised manuscript as follows:

“In contrast, calculations on the asymmetric hydrogenation of $\text{Cr}(\text{CO})_3$ -ligated 4-chlorobenzophenone revealed relatively lower energy barriers to the formation of both enantiomeric products, which can probably be attributed to the electron-withdrawing effect of the $\text{Cr}(\text{CO})_3$ unit (for atomic charge population and conceptual density functional analysis see Supplementary Table 3,4).”

Supplementary Table 3 Atomic charge population.

Entry	Atom	Hirshfeld Charge	ADCH Charge	CM5 Charge
1	C1 (1a)	0.1520	0.2304	0.1853

2	O1 (1a)	-0.2507	-0.3179	-0.3009
3	C1 (1a-Cr)	0.1591	0.2505	0.1910
4	O1 (1a-Cr)	-0.2382	-0.2961	-0.2894

Supplementary Table 4 Vertical electron affinity and localized Fukui indexes (f_+) of **1a** and **1a-Cr**.

Entry	Compound	VEA (eV)	f_+	
1	1a	0.6520	C1	0.0386
			O1	0.1225
2	1a-Cr	0.9468	C1	0.1019
			O1	0.1154

REVIEWERS' COMMENTS

Reviewer #1 (Remarks to the Author):

The resubmitted paper has been accordingly revised, and the authors addressed most of reviewers' comments. Regarding my request to demonstrate kinetic resolution, I additionally recommend to cite a pioneering work on the stereoselective transfer hydrogenation of (η^{6} -aryl ketone)chromium: C. V. Ursini, G. H. M. Dias, J. A. R. Rodrigues, *J. Organomet. Chem.* **2005**, *690*, 3176-3186.

Overall, I'm happy to support publication in Nature Communications.

Reviewer #2 (Remarks to the Author):

The authors have properly addressed the most relevant issues raised by this reviewer and I am very happy to recommend the publication of the manuscript in Nature Communications.

Reviewer #3 (Remarks to the Author):

The authors have addressed my concerns with the original manuscript. The revised manuscript is ready for publication.

Referee 1:

1. Comments: The resubmitted paper has been accordingly revised, and the authors addressed most of reviewers' comments. Regarding my request to demonstrate kinetic resolution, I additionally recommend to cite a pioneering work on the stereoselective transfer hydrogenation of (η^6 -aryl ketone)chromium: C. V. Ursini, G. H. M. Dias, J. A. R. Rodrigues, *J. Organomet. Chem.* **2005**, *690*, 3176-3186.

Overall, I'm happy to support publication in Nature Communications.

Response: We thank the reviewer for the suggestion. We have cited this pioneering work on the stereoselective transfer hydrogenation of (η^6 -aryl ketone)chromium (*J. Organomet. Chem.* **2005**, *690*, 3176-3186) in the revised manuscript as follows:

“87. Ursini, C. V., Dias, G. H. M. & Rodrigues, J. A. R. Ruthenium-catalyzed reduction of racemic tricarbonyl(η^6 -aryl ketone)chromium complexes using transfer hydrogenation: A simple alternative to the resolution of planar chiral organometallics. *J. Organomet. Chem.* **690**, 3176–3186 (2005).”